# Node2binary: Compact Graph Node Embeddings Using Binary Vectors

## Abstract

With the adoption of deep learning models to low-power, small-memory edge devices, energy consumption and storage usage of such models has become a key concern. The problem acerbates even further with ever-growing data and equally-matched bulkier models. This concern is particularly pronounced for graph data due to its quadratic storage, irregular (non-grid) geometry, and very large size. Typical graph data, such as road networks, infrastructure networks, social networks easily exceeds millions of nodes, and several gigabytes of storage is needed just to store the node embedding vectors, let alone the model parameters. In recent years, the memory issue has been addressed by moving away from memory-intensive double precision floating-point arithmetic towards single-precision or even half-precision, often by trading-off marginally small performance. Along this effort, we propose Node2binary, which embeds graph nodes in as low as 128 binary bits, which drastically reduces the memory footprint of vertex embedding vectors by several order of magnitude. Node2binary leverages a fast community detection algorithm to covert the given graph into a hierarchical partition tree and then find embedding of graph vertices in binary space by solving a combinatorial optimization (CO) task over the tree edges. CO is NP-hard, but Node2binary uses an innovative combination of discrete gradient descent and randomization to solve this effectively and efficiently. Our extensive experiments over four real-world graphs show that Node2binary achieves competitive performances compared to the state-of-the art graph embedding methods in both node classification and link prediction tasks.

## CCS Concepts

• **Mathematics of computing** → **Combinatorial optimization**; • **Computing methodologies** → **Discrete space search**; **Randomized search**; **Ontology engineering**.

## Keywords

Binary Space Embedding, Graph Embedding, Discrete Gradient Descent, Randomized Algorithm

**ACM Reference Format:**
Anonymous Author(s). 2018. Node2binary: Compact Graph Node Embeddings Using Binary Vectors. In *Proceedings of Make sure to enter the correct conference title from your rights confirmation emai (Conference acronym 'XX).* ACM, New York, NY, USA, 10 pages. https://doi.org/XXXXXXX.XXXXXXX

## 1 Introduction

In today's connected world, networks have become a formidable data structure for representing many complex systems; examples include social networks, biological networks, transportation networks, information networks, etc. The information captured in such networks holds immense value, both commercial and strategic, so developing knowledge discovery models for analyzing these networks have received increasing attention by both researchers and practitioners. Since most of the knowledge discovery models are suited for a vector space, a fundamental task for supporting such analysis is to embed a network in a latent space in which each vertex is represented as a low-dimensional vector. This task is known as graph representation learning (GRL). A key objective of GRL is to find vertex representation (i.e., vertex embedding) in a way that the proximity (based on graph topology, or property) among the vertices are preserved in the embedding space. Such a low-dimensional embedding is very useful in a variety of applications, such as visualization [1, 17], vertex classification [10, 13], and link prediction [4].

Over the years, as real-life graphs grew larger, often exceeding millions of vertices, graph representation learning (GRL) task has become computationally challenging. Lately, the GRL task is mostly solved by graph neural networks or its variants, which are computationally expensive—this also contribute to the challenge. Besides computation, for large graphs the memory footprint of the embedding vectors is also large. Say, for a graph with 10 millions vertices, if each embedding vector are of 100 dimension (a typical number), the memory footprint of these vectors is more than 60 GB considering double precision real numbers. If single precision real numbers are used, the number reduces to 30 GB, still a formidable amount of storage. As machine learning tasks are now being solved in the edge devices, which are battery powered with limited memory storage; processing such large dataset is difficult due to memory bottleneck. If memory is not an issue, transferring such large amount of data to and from a server would drain the battery of such devices. Hence, there is a surge of interest in machine learning research to develop embedding methods that embed entities in fewer bits, resulting in efficient memory and storage requirements. However, there exist no prominent works for solving GRL that generate compact embedding vectors.

Over the last decade, plethora of methods have been proposed for solving the GRL task; prominent methods follow methodologies derived from matrix factorization [1, 9], structural-preserving optimization [12], node-context sampling using random walk [4, 10], and graph neural networks [17]. Some of these methods consider network topology, whereas a few others consider both topology and node/edge attributes. The matrix factorization based methods factor a matrix capturing node similarity; Laplacian matrix, its different variants, or specifically designed matrix capturing higher order node proximity are often used for this task [1]. Such method are

costly and generally not scalable for very large networks. To overcome the lack of scalability issue, random walk based methods [4] are proposed. These methods perform random walk sampling from each node to build a node-sequence capturing the context of the given node. Node embedding vectors are then learnt by maximizing the probability of node co-occurrences in the generated node-sequence. Performance of such model is highly dependent on the quality of sampling strategy which generates the node-context; besides, a large number of samples are needed for capturing the node context effectively. Furthermore, it has been shown that random walk-based embedding approaches implicitly perform factorization of a properly chosen dense transition probability matrix, leading to better performance on downstream tasks [11]. Methods based on structure preservation learns node embedding by optimizing a loss function which directly captures first- or higher-order node similarity. But performance of such methods are generally not very good [13, 17].

With the popularity of deep learning, various graph neural network architectures have also been proposed for learning node embedding. The earliest among these is GCN (graph convolution network) [7], which uses graph convolution or neighborhood aggregation, a method for enriching node representation by merging its features with those of its neighbors. But GCN is a transductive model thus cannot be generalized to unseen graph nodes. Graph-SAGE [6] overcomes this limitation by efficiently using network topology and node attributes to generate embeddings for new nodes. While graph neural networks use latest technologies from the deep learning community for the task of GRL, there are two crucial limitations. First, mostly all of the architectures assumes the presence of node attributes, which is not the case for general node embedding task. Also, many of the models are trained in a supervised setup, but for a general GRL task, node/edge labels are not typically available.

In this work, we propose NODE2BINARY, a novel node embedding method which embeds vertices of a given graph using binary bitvectors. NODE2BINARY adopts an out-of-the-box idea for binary embedding which stems from a community centric viewpoint of the graph. NODE2BINARY first builds a community partition tree, a hierarchical clustering of the vertices of the input graph. Using this community-view of the graph, NODE2BINARY imposes constraints over the community partition tree edges to learn meaningful embedding of the graph vertices. Given that the embedding space for NODE2BINARY is binary, the learning task becomes a constrained combinatorial optimization, which NODE2BINARY solves by using a synthesis of discrete gradient descent and randomized local search. Extensive experimental results show the superiority of NODE2BINARY over a number of baseline graph embedding models.

Our contributions can be summarized as follows:

- We propose a novel combinatorial optimization framework for learning vertex representation vectors of an input graph in the binary space. The proposed optimization framework uses discrete gradient descent and randomization algorithm.
- Experiments on four real-world networks demonstrate that NODE2BINARY maintains competitive performance compared to the state-of-the-art methods, with superior performance at increasingly smaller number of bits.

## 2 Related Work

The main motivation of embedding in binary space is to have a compact node representation which is efficient both computationally and storage-wise. In some existing works, binary vectors are used to embed nodes of a general network, where the primary objective is to perform node similarity search by hashing binary vectors developed through fast sketching methods. The latest among these works is called NODESIG [23] which uses stable random projection for learning binary embedding of vertices. Most of the binary embedding works take inspiration from the random projection based fast nearest neighbor search using locality sensitive hashing [18, 19]. Our work is different from the existing works which uses hashing for embedding nodes in binary space. A recent work [5] embeds entities having is-a relation by using binary bitvectors. Thus it is suitable for embedding any directed network (say, tree or DAG) which satisfy transitivity property along the edge direction.

Several recent methods [2, 8] use graph coarsening on the original graph, then they apply the methodologies of a traditional graph-embedding method (such as node2vec [4], NetMF [11] etc.) on the coarsest graph to produce binary embeddings. Whereas in this work, we learn vertex embedding vectors by solving combinatorial optimization directly in the binary space. Some simple approaches utilize random projection [3, 22] and spectral graph sparsification techniques [21] to learn scalable network embeddings, but they sacrifice performance in doing so.

## 3 Methodology

**Notations:** $G$ is the input graph for which we are soliciting vertex representation vectors; $V$ is the vertex-set and $E$ is the edge set of $G$. NODE2BINARY builds a hierarchical partition tree of the vertices of $G$, denoted with the symbol $\mathcal{T}_G$. Italic letters $a$, $b$ are used to denote the vertices of the graph $G$ and also nodes of the tree $\mathcal{T}_G$. NODE2BINARY learns embedding vectors for each vertex of $G$ and also for each nodes of $\mathcal{T}_G$. To represent these embedding vectors we use boldface letters, $\mathbf{a}$ for node $a$. The letter $d$ is a positive integer number denoting the embedding dimension. Greek letters, such as $\alpha$, $\beta$, $\gamma$ are scalars. They are generally reserved for user-defined hyperparameters. The symbol $\Delta_{\mathbf{x}}$ is used for gradient with respect to a vector variable, $\mathbf{x}$.

### 3.1 Problem Formulation and Framework

Graph embedding in binary space is defined as: Given an undirected graph $G(V, E)$, learn an embedding function $f : V \rightarrow \{0, 1\}^d$ that embeds *similar* vertices in $V$ close to each other in the embedding space. The embedding space is discrete, as each graph node is mapped to the vertices of unit-length $d$-dimensional hypercube. We also assume that the node set $V$ and edge set $E$ do not have attributes, so two vertices in $G$ are considered similar if they appear within the same local context in the graph space. Embedding function $f$ must preserve this similarity in the embedding space, wherein similarity between two bitvectors can be defined by Hamming distance.

To solve this embedding task, NODE2BINARY uses an innovative combinatorial optimization framework. To set up this combinatorial optimization, NODE2BINARY first partitions the vertex-set of $G$ into hierarchical disjoint communities, which can be organized into a community partition tree. Then the binary embedding vectors

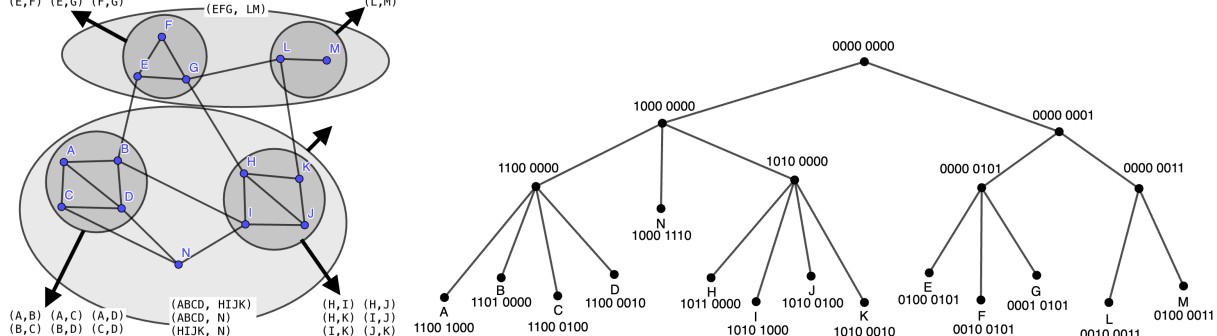

**Figure 1: *Hierarchy Tree* formation from the Original Graph. (Spaces in binary embeddings are for ease of reading.)**

of the vertices of $G$ are then learnt by respecting the following two requirements: (1) In a hierarchical (tree-like) partitioning of a network into communities, where the root node of the partitioning is the entire graph, and each of the leaf nodes is a single vertex, embedding vectors of the vertices is more similar to that of the parent community than that of any of the ancestral community; (2) If two vertices are part of the same community, their embedding vectors are similar. Given that the embedding space is binary, satisfying these requirements leads to a combinatorial optimization problem, which Node2binary solves by using an innovative combination of discrete gradient descent and randomization. We discuss the overall methods of Node2binary in the following subsections.

## 3.2 Constructing Community Partition Tree

We want the embedding vectors of two vertices to have small hamming distance, if the vertices are well-connected. To realize the notion of well-connectedness between two vertices, we adopt a community-centric view of the graph, where the graph is hierarchically partitioned into successively smaller communities. This partitioning can also be shown by a tree often known as dendogram in hierarchical clustering literature. We refer this tree as *community partition tree*. For a graph $G(V, E)$, the community partition tree is denoted as $\mathcal{T}_G$. Nodes in $\mathcal{T}_G$ represents a community and an edge from a parent to a child in $\mathcal{T}_G$ represents community to sub-community relation. Note that in this community-hierarchy, a parent community is partitioned into disjoint children community recursively until a community contain only a single vertex of $G$. Thus the root node of $\mathcal{T}_G$ represents a community containing all the vertices in $V$, and each of the leaf nodes of $\mathcal{T}_G$ is one of the vertices in $V$. For a toy example, see Figure 1. On the left side of this figure, we show a graph in its hierarchically-partitioned form. The graph is initially partitioned into two communities, each of which are partitioned further into smaller communities. The community partition tree of this graph is shown on the right side of the figure.

In such a partitioning if two vertices belong to the same community, they are considered well-connected and should have similar bitvector representation. We also extend the idea of similarity between vertices to similarity between communities and to achieve that we assign bitvector representation for the communities (the nodes in the community partition tree), as well. This is illustrated in Figure 1, where each of the tree node (in the right side of the figure) is assigned a bitvector. Then the embedding task becomes

assigning a binary vectors to each of the nodes of the community partition tree satisfying certain requirements (which is discussed in the next subsection). We use Leiden algorithm [16] to construct the community partition tree ($\mathcal{T}_G$) from the given graph ($G$). The algorithm is applied recursively to generate community partition hierarchy. The leaf nodes correspond to vertices in the graph $G$, and internal nodes correspond to communities ordered by inclusion. We summarize the steps of the formation of our community partition tree in Algorithm 1.

---

**Algorithm 1** CreateTree

**Require:** Graph $G = (V, E)$, number of layers $\ell$
**Ensure:** Hierarchy tree $\mathcal{T}_G$, whose elements are set of vertices
 1: $\mathcal{T}_G \leftarrow$ single-node tree with node $V$
 2: **if** $\ell = 1$ or $|V| = 1$ **then**
 3:     Append child $\{v\}$ to $\mathcal{T}_G$, for each $v \in V$
 4: **else**
 5:     $C \leftarrow \text{Leiden}(G)$
 6:     $\mathcal{T}_G \leftarrow \{\}$
 7:     **for** community $C \in \mathcal{C}$ **do**
 8:         $G_C \leftarrow$ induced subgraph of $C$
 9:         Append child CreateTree($G_C, \ell - 1$) to $\mathcal{T}_G$
10:     **end for**
11: **end if**

---

## 3.3 Mathematical Framework of Node2binary

To map all the vertices which belong to a community to similar bitvectors, we ensure that all vertices in a community adhere to certain *behaviors*. To realize this notion, we utilize bitvectors assigned to each community. As stated earlier, each node in the hierarchical partition tree ($\mathcal{T}_G$) is assigned a bitvector. If we hypothesize that bitvector indices are indicator functions of possessing a given behavior, for a representation vector of a community $C$, the '1' indices denote that the members of $C$ adhere to those behaviors. If $C$'s members are partitioned into sub-communities, sub-communities members also possess those behaviors inherited from their parent ($C$). This requirement extends to the entire partition hierarchy; i.e., if a community possess a behavior ('1' bit), all its sub-communities will also possess the same behavior, reflected in their bitvector representatioin. In this way, representation vector of every node in $\mathcal{T}_G$ inherits the '1' bits from its parents, and may add '1' bit(s) of its own.

The terminal nodes in $\mathcal{T}_G$ are vertices of the graph $G$, hence the bitvectors of these nodes also follow this requirement. For illustration, see the community partition tree in Figure 1. In this tree, the leaf node $H$ (which is also a vertex of the input graph) is assigned an embedding vector 10110000; there are 3 '1' bits, of which two (the first and the third indices) are inherited from its parent, and the last (the fourth index) one is of its own. Given a hierarchical partition tree, the embedding model of NODE2BINARY solves a constrained combinatorial optimization problem for assigning the bit vectors of each node in this tree. If node $a$ is a child of node $b$ in $\mathcal{T}_G$, then $f(b)[i] \Rightarrow f(a)[i], \forall i \in [1 : d]$, where $d$ is the embedding dimension and the symbol '$\Rightarrow$' is logical implication. This can be succinctly shown as $\mathbf{b} \Rightarrow \mathbf{a}$, where $\mathbf{b} = f(b)$ and $\mathbf{a} = f(a)$.

The above requirement leads to a combinatorial feasibility problem over the nodes of the community partition tree $\mathcal{T}_G$ of a given graph $G$. The number of constraints equals the number of edges in $\mathcal{T}_G$ times the dimension, as we need to satisfy the implication condition for each tree edge and for each index of the embedding vectors. A solution to this problem is a $d$-dimensional bitvector for each node of the partition tree so that the parent to child implication discussed above is satisfied. Solving this problem exactly is NP-Hard, so we convert it to an unconstrained optimization problem with a loss function equal to the number of unsatisfied constraints. For a given partition tree and $d$, there exist many feasible solutions for the above problem. To narrow the solution space, we include preferences over the solutions via sibling similarity.

If $c$ and $d$ are children of a node $b$ in $\mathcal{T}_G$, then embedding vectors of $c$ and $d$ inherits the '1' bits of $b$; this makes $c$ and $d$ similar, but if $b$ does not have many 1 bits, the constraint makes a very weak similarity. So, we enforce sibling similarity in the objective function to prefer that $c$ and $d$ are also similar in bit indices where $b$ has a '0' bit. However, this requirement is not a constraint as we do not want $c$ and $d$ to have identical embedding vectors. So, for each node $b \in \mathcal{T}_G$, we take $k$ random sample of its children and build a loss function to minimize Hamming distance between pairs of children.

### 3.4 Training Algorithm

Our training goal is twofold. The first is to ensure that *tree relation: parent-child* on nodes (including internal nodes) of $\mathcal{T}_G$ maps to *bit-wise implication* in the embedding space, and the second is to ensure that *sibling nodes* in $\mathcal{T}_G$ have *similar* embedding vectors as defined by Hamming distance.

Our *loss function* is defined as follows. Let $H$ be the set of *hierarchy pairs* of nodes $(a, b)$ from $\mathcal{T}_G$ satisfying $b \Rightarrow a$ ($a$ is child and $b$ is parent node), and $S$ the set of *sibling pairs* $(c, d)$ in $\mathcal{T}_G$. We have two goals: ensure that (1) nodes in the *hierarchy* pass their "1" bits down to their children, and (2) two *sibling* nodes have embeddings with small Hamming distance between them. Condition (1) is equivalent to ensuring that, since node $a$ is a child of $b$, $\mathbf{a}$ inherits all 1 bits from $\mathbf{b}$, i.e. there is no $j$ such that $\mathbf{a}_j = 0$ and $\mathbf{b}_j = 1$. Let $H$ be the set of all pairs $(a, b)$ such that $b$ is an ancestor of $a$. We split this condition into *positive* loss, for $(a, b) \in H$, and use *negative sampling* as in to compute an additional loss function for the negative pairs.

$$\text{Loss} = \alpha\text{Loss}^{\text{hier},+} + \beta\text{Loss}^{\text{hier},-} + \gamma\text{Loss}^{\text{sib}} \quad (1)$$

$$\text{Loss}^{\text{hier},+} = \sum_{(a,b)\in H} \sum_{j=1}^{d} \left(1 \text{ if } (\mathbf{a}_j, \mathbf{b}_j) = (0, 1) \text{ else } 0\right) \quad (2)$$

$$\text{Loss}^{\text{hier},-} = |W|, \text{ where } W = \left\{(a, b) \notin H : \mathbf{a}_j \geq \mathbf{b}_j, \forall j\right\} \quad (3)$$

$$\text{Loss}^{\text{sib}} = \sum_{(a,b)\in S} \text{HammingDistance}(\mathbf{a}, \mathbf{b}) \quad (4)$$

### 3.5 Discrete Gradient using Boolean Logic

Once the *Hierarchy Tree* is formed and the *sibling pairs* are collected, we train our model to learn binary vector embeddings by using *discrete gradients*, which is inspired by the continuous Stochastic Gradient Descent (SGD) method. If $f(\mathbf{x})$ is a function whose inputs are binary, that is, $\mathbf{x} \in \{0, 1\}^k$, then the *discrete gradient* of $f$ with respect to $\mathbf{x}$ can be defined as

$$(\Delta_{\mathbf{x}} f(\mathbf{x}))_j = f(\mathbf{x} \text{ with bit } j \text{ flipped}) - f(\mathbf{x}) \quad (5)$$

This differs from the standard definition of the derivative for a continuous function, in that our discrete gradient $\Delta_x f$ is positive when *changing* the binary $x$ increases $f$. We define the *discrete gradients* for the hierarchical tree structure, where node $a$ is a child of $b$, as

$$\Delta_{\mathbf{a}}\text{Loss}^{\text{hier},+} = \sum_{b:(a,b)\in H} -\mathbf{b} * (1 - 2\mathbf{a}) \quad (6)$$

$$\Delta_{\mathbf{b}}\text{Loss}^{\text{hier},+} = \sum_{a:(a,b)\in H} -(1 - \mathbf{a}) * (2\mathbf{b} - 1) \quad (7)$$

These equations have the property that $\Delta_{\mathbf{a}}\text{Loss}^{\text{hier},+}(\mathbf{a}, \mathbf{b})$ is $-1$ in position $j$ iff $(\mathbf{a}_j, \mathbf{b}_j) = (0, 1)$. This encourages the model to invert $\mathbf{a}_j$, thus restoring the partial order relation. Contrariwise, if $(\mathbf{a}_j, \mathbf{b}_j) = (1, 1)$, the gradient of $\mathbf{a}_j$ is set to $+1$, which discourages $\mathbf{a}_j$ from being flipped. A similar logic applies to the $\mathbf{b}$ gradient; we expand on this logic in Appendix A.2.

Similarly, we define the Hamming-loss gradient, which is the same for both inputs,

$$\Delta_{\mathbf{a} \text{ or } \mathbf{b}}\text{Loss}^{\text{sib}} = 1 - 2(\mathbf{a} \oplus \mathbf{b}) \quad (8)$$

The reader can verify that this vector is $+1$ in positions where $\mathbf{a}, \mathbf{b}$ agree, and $-1$ where they disagree. This encourages nodes in the same hierarchy to have similar embeddings.

With discrete space, we cannot adjust the embeddings by a "small amount" as in traditional gradient descent. Instead, we compute a probability for inverting bit $\mathbf{a}_j$ based on its overall gradient $\Delta(\mathbf{a}_j)$, FlipProb($\Delta(\mathbf{a}_j)$), defined as

$$\text{FlipProb}(x) = \frac{1}{2}\tanh\left(-2(r_\ell x + b_\ell)\right) \quad (9)$$

where the *learning rate* $r_\ell$ controls the flipping rate and the *bias* $b_\ell$ allows flipping bits with zero gradient, to avoid local maxima. This function ensures that at most half of all bits are flipped each iteration, which prevents the model from oscillating.

NODE2BINARY's algorithm is summarized in Algorithm 2. We start by building the tree as in Algorithm 1 and generating the sibling pairs $S$ and hierarchy pairs $H$. We initialize embeddings to the zero matrix. On each training iteration, we compute the discrete

**Table 1: Statistics of the datasets used in the experiments.**

| Dataset | PPI | DBLP | Blog | YouTube (Small / Large) |
|---|---|---|---|---|
| #V | 3,890 | 13,326 | 10,312 | 31,703 / 1,138,499 |
| #E | 38,705 | 34,281 | 333,983 | 276,126 / 2,990,443 |
| #L | 50 | 2 | 39 | 47 |

gradient matrix $\Delta \in \mathbb{Z}^{n \times d}$, pass $\Delta$ into FlipProb to get a probability matrix $P$, and flip each bit $\mathbf{a}_j$ with probability $P(a, j)$. The symbol Emb denotes the learned embedding function $\text{Emb} : \mathcal{T}_G \to \{0, 1\}^d$.

---

**Algorithm 2** Training Algorithm

---

**Require:** Graph $G = (V, E)$, dimension $d$, iterations $t$, parameters
    $(\alpha, \beta, \gamma, r_\ell, b_\ell)$
1: $\mathcal{T}_G \leftarrow \text{CreateTree}(G)$
2: $S \leftarrow \text{CreateSiblings}(\mathcal{T}_G)$
3: $H \leftarrow \{(a, b) : a, b \in \mathcal{T}_G, a \subset b\}$      ▷ $a, b$ are sets of vertices
4: $\text{Emb}(a) \leftarrow (0, \ldots, 0)$ for all $a \in \mathcal{T}_G$
5: **for** $\tau = 1$ to $t$ **do**
6:     $H^- \leftarrow$ negative samples of $H$
7:     $\Delta \leftarrow \alpha\Delta^{\text{hier},+}(H, \text{Emb}) - \beta\Delta^{\text{hier},-}(H^-, \text{Emb}) + \gamma\Delta^{\text{sib}}(S, \text{Emb})$
8:     $P \leftarrow \text{FlipProb}(\Delta)$
9:     **for** $a \in \mathcal{T}_G$ **do**
10:         $\text{Emb}(a) \leftarrow \text{Emb}(a) \oplus (\text{random}(0, 1) < P(a, \cdot))$
11:     **end for**
12: **end for**

---

## 3.6 Time Complexity and Space Efficiency

Our algorithm is very efficient in both the time and space dimensions. Algorithm 1 can be seen to take $O(\ell|E|)$ time. It outputs a tree with at most $\ell|V|$ nodes. Sibling sampling selects $k$ random siblings of each tree node, which is $O(k\ell|V|)$. Finally, the training algorithm is $O(t(|H| + |S|))$; $|H|$ is bounded by $\ell^2|V|$, as each of the $\ell|V|$ nodes have at most $\ell$ ancestors, and $|S| \in O(k\ell|V|)$ from before. Thus, node2binary runs in linear time.

For space complexity, the storage needed for $n$ nodes each having $d$-bit vectors is $\frac{dn}{8}$ bytes, while double- and single-precision floating point based numbers take $8dn$ and $4dn$ bytes, respectively. Thus, node2binary embeddings are 32 or 64 times more space efficient at the same dimension than most competing models.

## 4 Experiments and Results

We evaluate Node2binary's embedding quality based on two standard graph representation learning tasks: multi-label classification on vertices and link prediction on edges. We organize this section in the following way: In §4.1, we provide details of the datasets we have used. In §4.2, we discuss the baseline methods we compare our method with, and the evaluation metrics we used. In §4.3 we provide details of our experimental setup. In §4.4 and §4.5, we further discuss the experimental setup and results from our multi-label node classification and the link prediction experiments, respectively. For the remaining of the section, we discuss

the robustness of our method. In §4.6, we compare our method's scalability with state-of-the-art methods. In §4.7, we show how tree loss and sibling loss converge over time. In §4.8, we discuss parameter sensitivity, and in Appendix §A.4, we provide an ablation study to show how different components of our objective function affect our performance. Our code is publicly available at https://anonymous.4open.science/r/node2binary-C952/.

### 4.1 Datasets

For our experiments we have chosen four moderate to large size real world labeled graphs drawn from biological, collaboration and social network domains which are largely used by our competitors. **PPI** [4] is a subgraph of the protein-protein interaction network for *Homo sapiens*. The labels of the nodes represent its gene sets and also biological states. **DBLP** [20] is a collaboration network which captures the co-authorship of authors. The labels of a node in the co-author graph represents publication venues of the respective author. **BlogCatalog** [14] is a social network of bloggers where labels indicate the topics of interest by the corresponding blogger. Finally, **YouTube** [15] is a social network of users and labels refer to list of subscriptions (such as technology) by the user. We consider a subset of YouTube dataset for the node classification and link prediction whereas use the full dataset with millions of nodes for the scalability experiment. The statistics of the datasets are summarized in Table 1. All our datasets are unweighted and undirected.

### 4.2 Baseline Methods and Evaluation Metrics

We compare our method against seven carefully chosen baseline models. We can categorize our baseline models as the following: (1) random-walk based models: DeepWalk [10] and node2vec [4], (2) Neural Network based model: LINE [13], (3) matrix-factorization based model: HOPE [9], (4) Hash based model NodeSketch [19], (5) Inductive model GraphSAGE [6] with Mean aggregation (we use node degrees as features), and (6) Random projection binary model NODESIG [23], our main competitor. We omitted traditional methods like GraRep [1] and SDNE [17] and chose HOPE and LINE from those categories because of their superior performance.

For our experiments, we used the setup from [4] and considered the commonly used evaluation metrics by our competitors. For multi-label node classification task, we consider Macro-F1 score to show results across different embedding dimensions. We use AUC-ROC score for link prediction experiment.

### 4.3 Experimental Setup

We design experiments to evaluate our and the competitors performance per bits. Most of our floating-point based competitors are based on double-precision except LINE model which is based on single precision. NodeSketch uses integers so it uses 32 bits per dimension and NODESIG is a binary embedding method so it allocates a single bit per dimension. To be fair with all the models we considered dim $d$ as in the bit-range [128, 256, 512, 1024, 2048, 4096, 8192]. For Node2binary, we run 1000-epoch experiments, evaluating performance every 100 epochs. Since our algorithm is randomized, we repeated each experiment 5 times for each task and dataset. We reported the best results along with mean and standard deviation in Tables 2 and 3. Other than dimension $d$, we have hyper

**Table 2: Multi-label Node Classification task, according to number of *bits* in embeddings**

| | DeepWalk | node2vec | LINE | HOPE | NodeSketch | GraphSAGE | NODESIG | node2binary |
|---|---|---|---|---|---|---|---|---|
| bits (dim) | | | | | | | | |
| **YouTube** | | | | | | | | |
| 128 | 6.01 | 5.94 | 6.0 | 5.92 | 6.69 | 6.54 | **8.97** | 8.79 (8.69 ± 0.07) |
| 256 | 6.17 | 6.17 | 6.23 | 6.02 | 6.85 | 6.52 | 9.09 | **9.27** (9.22 ± 0.06) |
| 512 | 6.44 | 6.39 | 6.52 | 6.26 | 7.12 | 6.54 | 8.75 | **9.51** (9.44 ± 0.06) |
| 1024 | 7.0 | 7.38 | 6.63 | 6.55 | 7.45 | 6.58 | 8.38 | **9.51** (9.47 ± 0.03) |
| 2048 | 7.92 | 8.18 | 7.38 | 6.77 | 8.27 | 6.55 | 8.87 | **9.5** (9.48 ± 0.02) |
| 4096 | 8.89 | 8.86 | 8.1 | 7.89 | 8.48 | 6.55 | 9.5 | **9.67** (9.52 ± 0.09) |
| 8192 | 9.4 | 9.35 | 9.3 | 8.42 | 8.85 | 6.55 | **9.93** | 9.56 (9.45 ± 0.09) |
| **DBLP** | | | | | | | | |
| 128 | **82.3** | 73.43 | 45.77 | 39.03 | 41.89 | 39.26 | 73.38 | 76.44 (73.68 ± 2) |
| 256 | **88.52** | 86.39 | 48.98 | 40.66 | 44.13 | 39.26 | 79.14 | 82.87 (81.91 ± 0.8) |
| 512 | 89.16 | 87.31 | 56.95 | 51.21 | 47.5 | 48.73 | 84.05 | **94.38** (94.25 ± 0.1) |
| 1024 | 89.19 | 88.45 | 55.20 | 51.63 | 48.61 | 48.69 | 87.32 | **95.71** (95.46 ± 0.2) |
| 2048 | 89.87 | 89.47 | 60.42 | 62.36 | 53.01 | 48.73 | 93.47 | **95.82** (95.5 ± 0.2) |
| 4096 | 90.26 | 89.38 | 72.08 | 70.17 | 55.95 | 48.7 | **96.62** | 94.79 (94.77 ± 0.01) |
| 8192 | 93.73 | 89.93 | 79.08 | 75.11 | 59.45 | 48.7 | **97.6** | 94.81 (94.79 ± 0.02) |
| **Blogcatalog** | | | | | | | | |
| 128 | 3.12 | 3.27 | 3.28 | 2.83 | 2.83 | 2.96 | 11.85 | **12.1** (11.83 ± 0.2) |
| 256 | 6.27 | 6.98 | 3.46 | 2.90 | 3.81 | 2.57 | **15.29** | 14.64 (14.38 ± 0.1) |
| 512 | 10.89 | 12.23 | 3.90 | 3.33 | 4.45 | 2.84 | **16.1** | 16.08 (15.61 ± 0.4) |
| 1024 | 17.19 | 18.49 | 5.74 | 3.71 | 4.98 | 2.56 | **18.73** | 15.49 (15.49 ± 0.3) |
| 2048 | 22.34 | **23.43** | 10.51 | 7.05 | 6.73 | 2.56 | 22.06 | 15.81 (15.5 ± 0.3) |
| 4096 | 24.69 | **26.07** | 22.42 | 11.42 | 8.23 | 2.56 | 22.03 | 13.95 (13.86 ± 0.06) |
| 8192 | 25.49 | **26.18** | 25.41 | 13.95 | 9.63 | 2.56 | 24.27 | 13.95 (13.93 ± 0.02) |
| **PPI** | | | | | | | | |
| 128 | 4.19 | 6.07 | 3.73 | 3.92 | 3.48 | 1.57 | 13.44 | **16.89** (16.55 ± 0.3) |
| 256 | 8.73 | 9.73 | 3.59 | 5.85 | 4.51 | 1.57 | 13.21 | **17.93** (17.48 ± 0.4) |
| 512 | 13.03 | 13.25 | 3.87 | 7.03 | 5.17 | 1.55 | 13.74 | **17.86** (17.21 ± 0.4) |
| 1024 | 16.34 | 17.21 | 5.28 | 9.14 | 5.90 | 2.76 | 16.06 | **17.73** (17.13 ± 0.4) |
| 2048 | 18.04 | **18.16** | 7.61 | 11.26 | 6.99 | 1.55 | 17.96 | 17.07 (16.95 ± 0.1) |
| 4096 | 18.7 | **19.12** | 16.24 | 12.35 | 7.07 | 1.55 | 18.93 | 16.4 (16.27 ± 0.09) |
| 8192 | 18.95 | 18.29 | 18.72 | 13.96 | 7.32 | 1.55 | **20.96** | 16.42 (16.23 ± 0.12) |

Models — Node Classification (Metric: F1 %)

parameters: depth of the hierarchy tree $l$ and sibling similarity coefficient $\gamma$. Hyper parameters for the discrete gradient computations are positive and negative sample weights $\alpha$ and $\beta$, negative sample multiplier $n^-$, learning rate $r_l$ and bias $b_l$ to get binary embedding. We randomly sample $k = 10$ siblings per entity for all of our experiments. We reuse community partition tree across different dimension once it is formed, to ensure consistency of results. Apart from the dimension $d$, we used best reported hyper parameters in our competitors works. Following [4], we learned best in-out $p$ and return $q$ hyper parameters by 10 fold cross-validation on 10% labeled data using a grid search over 0.25, 0.5, 1,2,4. We ran all the models on a Tesla A100 GPU with 128 GB memory.

## 4.4 Multi-label Node Classification

**Experimental setting:** Node classification task utilizes a *labeled* dataset, where each node of the graph has one or more labels from a labelset $L$. The task is to correctly classify all the labels for each node. The task becomes more challenging as the size of $L$ increases. Once we have the embeddings for each node in the dataset, we perform 10-fold cross-validation and randomly sample 50% of nodes to train a OnevsRestClassifier model with Logistic Regression using 'Liblinear' solver for 1000 iterations, while keeping the other 50% nodes to evaluate classification performance. We also randomly subsample 50% nodes from the YouTube dataset before the classification task because of its larger size than other datasets.

**Experimental results:** We report our node classification experimental results in Table 2 with a decreasing order of datasets based on their node counts. Based on the results we are either best (bold) or second best (underline) most of the time, particularly as the number of bits decrease. For the largest YouTube dataset we reach 9.27% F1-score at just 256 bits whereas most of our competitors (except NODESIG) need 8192 bits to achieve similar performance. For the second largest DBLP dataset, we quickly achieve 94% F1-score at 512 bits and maintain superior performance across all dimensions. DBLP is a 2-labeled dataset, so most competitors perform better on it. For BlogCatalog dataset again we perform competitively at lower-bit resolution. The closest competitor to Node2binary is another binary model, NODESIG, which performs well at low bit resolution.

Table 3: Link Prediction task, according to number of *bits* in embeddings

| bits (dim) | DeepWalk | node2vec | LINE | HOPE | NodeSketch | GraphSAGE | NODESIG | node2binary |
|---|---|---|---|---|---|---|---|---|
| | | | | | **Models** | | | |
| | | | | **Link Prediction (Metric: AUC %)** | | | | |
| | | | | | **YouTube** | | | |
| 128 | 55.31 | 55.56 | 60.03 | 52.44 | 56.22 | 50 | 69.54 | **75.62** (73.39 ± 1.31) |
| 256 | 44.07 | 46.81 | 64.4 | 65.19 | 60.66 | 65.77 | 74.36 | **78.31** (76.6 ± 1.03) |
| 512 | 56.43 | 58.96 | 71.31 | 63.1 | 68.27 | 50 | 75.44 | **80.49** (79.87 ± 0.43) |
| 1024 | 62.35 | 65.01 | 71.58 | 63.02 | 66.62 | 65.77 | 79.62 | **83.16** (82.39 ± 0.57) |
| 2048 | 63.16 | 64.37 | 75.06 | 62.51 | 71.59 | 65.77 | 82.6 | **85.08** (84.48 ± 0.33) |
| 4096 | 64.24 | 69.45 | 73.65 | 62.89 | 70.65 | 65.77 | 83.19 | **85.52** (85.23 ± 0.22) |
| 8192 | 63.56 | 68.89 | 76.31 | 61.91 | 73.46 | 65.77 | **86.79** | 84.53 (84.3 ± 0.14) |
| | | | | | **Blogcatalog** | | | |
| 128 | 49.05 | 46.31 | 57.48 | 53.06 | 51.96 | 50.5 | 58.93 | **76.8** (74.92 ± 1.22) |
| 256 | 60.86 | 59.06 | 56.81 | 55.09 | 52.91 | 50 | 63.01 | **80.27** (77.93 ± 1.58) |
| 512 | 61.61 | 59.68 | 62.8 | 52.39 | 56.09 | 50.5 | 65.89 | **80.42** (79.38 ± 0.58) |
| 1024 | 62.83 | 59.5 | 66.68 | 53.35 | 50.86 | 50.5 | 61.64 | **80.55** (79.98 ± 0.63) |
| 2048 | 59.84 | 60.57 | 69.06 | 53.31 | 59.55 | 50.5 | 67.35 | **80.08** (79.49 ± 0.44) |
| 4096 | 62.0 | 60.79 | 71.29 | 53.5 | 61.71 | 50.5 | 78.47 | **79.24** (78.82 ± 0.26) |
| 8192 | 58.51 | 60.38 | 69.96 | 53.59 | 62.76 | 50.5 | **83.48** | 77.48 (77.02 ± 0.28) |
| | | | | | **PPI** | | | |
| 128 | 51.77 | 51.66 | 49.17 | 48.48 | **63.1** | 50 | 51.74 | 57.35 (56.86 ± 0.49) |
| 256 | 53.74 | 53.75 | 50.28 | 49.3 | **62.53** | 50 | 52.9 | 59.44 (58.3 ± 0.95) |
| 512 | 54.24 | 52.82 | 51.05 | 49.47 | 62.64 | 50.42 | 58.75 | **63.8** (62.28 ± 1.05) |
| 1024 | 54.72 | 53.28 | 52.71 | 50.68 | 60.64 | 50.42 | 59.54 | **63.27** (62.78 ± 0.4) |
| 2048 | 51.31 | 53.99 | 58.58 | 50.13 | 62.49 | 50.42 | **67.38** | 63.99 (63.41 ± 0.41) |
| 4096 | 52.66 | 52.44 | 53.7 | 50.05 | 64.11 | 50.42 | **70.53** | 62.91 (62.51 ± 0.35) |
| 8192 | 51.67 | 51.8 | 53.9 | 50.33 | 62.7 | 50.42 | **70.44** | 61.35 (60.94 ± 0.41) |
| | | | | | **DBLP** | | | |
| 128 | 47.9 | 47.3 | 63.51 | 53.24 | **66.34** | 50 | 54.61 | 54.49 (53.56 ± 0.76) |
| 256 | 46.68 | 47.9 | 63.53 | 52.93 | **67.32** | 50 | 55.47 | 55.69 (54.72 ± 0.67) |
| 512 | 49.98 | 49.9 | 62.92 | 53.32 | **68.79** | 51.76 | 57.44 | 56.69 (56.11 ± 0.44) |
| 1024 | 48.69 | 49.04 | 62.0 | 53.43 | **64.33** | 51.76 | 58.3 | 58.4 (58.02 ± 0.29) |
| 2048 | 48.99 | 51.23 | 59.4 | 53.4 | **64.36** | 51.76 | 59.21 | 60.59 (60.32 ± 0.31) |
| 4096 | 49.96 | 51.27 | 56.85 | 51.99 | **62.62** | 51.76 | 61.51 | 60.78 (60.61 ± 0.2) |
| 8192 | 50.91 | 52.17 | 54.78 | 51.12 | **61.27** | 51.76 | 60.97 | 59.19 (58.66 ± 0.35) |

Among other competitors DeepWalk and node2vec perform best or second best for this dataset at higher bit-resolutions. For PPI dataset we achieve 18% F1-score at 256 bits unlike our major competitors who require 2048 bits to achieve similiar F1-score. The biggest take-away from this experiment is that most of our competitors need many bits to perform well, but Node2binary is competitive with significantly fewer bits. Even binary-based NODESIG performs worse than our model at low dimensions. For a finer-grained analysis of node classification, we vary train-test ratio from 0.1 to 0.9 keeping $d = 512$ and rest of the experimental setup as before. We report these results in Appendix A.3.

## 4.5 Link Prediction

**Experimental setting:** Link prediction is the task of determining, given two nodes in a graph, whether or not there is an edge (link) between them. We perform 'Hadamard' operation to get an edge embedding vector from the constituting node embedding vectors. We randomly sample 50% edges in the graph and use it as train set, ensuring that the train subgraph is connected. We also generate a collection of negative samples from both train and test set respectively by taking node pairs without an edge. We train a Logistic Regression classifier using edge embeddings from the train dataset as features with 'liblinear' solver with max iterations at 1000. For the Blogcatalog and Youtube datasets, we subsample 50% of nodes ensuring graph connectivity from the original graphs *before* the train-test split due to their larger edge counts than other datasets. After subsampling, BlogCatalog has 12.7% and YouTube has 15.67% of their original edge counts.

**Experimental results:** We report link prediction experimental results in Table 3 with a decreasing order of datasets based on edge counts. We perform better for the largest YouTube and BlogCatalog datasets across all bit-resolutions, except at 8192 bits where we lose to our binary-based competitor NODESIG. For BlogCatalog dataset, we achieve 80% AUC score at just 256-bits, whereas NODESIG requires 8192-bits. Most of our competitors could not cross 70% AUC-score for this dataset. For PPI, we come out best or second best at lower-bit resolutions losing to NodeSketch and NODESIG at high bit resolutions. These results again show that our binary

**Table 4: Algorithm Running Time on YouTube dataset (Task: Link Prediction)**

| Model | NodeSketch | NODESIG | NODE2BINARY |
|-------|-----------|---------|-------------|
| Time (s) | 2488 | 6727 | Hier Tree = 184 |
| | | | Alg Run = 5839 |
| Speed Up | 0.41x | 1.12x | 1x |

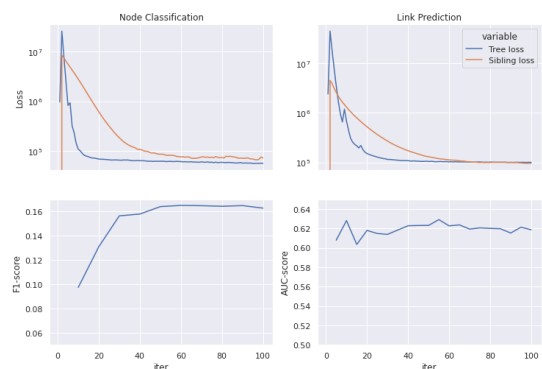

**Figure 2:** *Convergence* results (first 100 iterations) for both tasks on PPI dataset setting embedding dimension at 512.

competitor NODESIG requires large bit-resolution to perform well, unlike us. For the smallest dataset DBLP, NodeSketch does best across all dimensions. Our model overall came out third best after LINE across all dimensions. GraphSAGE performs poorly across all datasets due to lack of built-in node attributes.

### 4.6 Scalability

**Experimental setting:** We compare our algorithm training time with the competitors for the large YouTube dateset (Node count 1.1M, Edge count: 3M). After forming community partition tree, for hierarchical pairs we randomly sample indirect edges to keep total edge counts $\leq$ 2M and total sibling pairs $\leq$ 1M. We keep node2binary at dimension 512 and run for 100 iterations. We repeat this experiment 5 times and take average time. We give our competitors the best hyper parameters from their papers. We omit comparison with random-based, matrix-factorization, neural-network and inductive models since they take much longer to run.

**Experimental Results:** Among the competitors NodeSketch takes about 0.41x time, and NODESIG 1.12x, compared to our model. Our implementation is in Python, whereas our scalable competitors implemented their models either in C or MATLAB which are significantly faster programming languages. We will implement our model using a faster language in a future work.

### 4.7 NODE2BINARY Model Convergence Results

To justify our model convergence, we plot both Hierarchy Tree loss and Sibling Similarity loss for 100 iterations along with how evaluation metric changes for both node classification and link prediction tasks. For this experiment we use PPI dataset with dimension 512. We show our results in Fig 2. Since each loss starts at zero, there is a sudden jump initially and then exponential decay as expected. For both tasks, as tree loss and sibling similarity loss converges to a minima, each evaluation metric saturates to a maximum value.

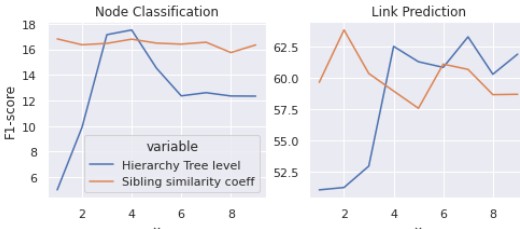

**Figure 3:** *Parameter sensitivity* experiment by varying tree level and sibling similarity coefficient on PPI dataset for embedding dimension set at 512.

### 4.8 Parameter Sensitivity

We mentioned the hyperparameters used for our model in the Experimental Setup section. Two hyperparameters are critical for NODE2BINARY's performance, so we elaborate their effect on node classification and link prediction performance using PPI dataset.

**Effect of Hierarchy Tree level, $l$:** CreateTree is an important step for our method. We use Leiden algorithm to partition the graph and repeat the process for $l$ layers, where $l$ is a hyperparameter. Our experimental section shows that for node classification performance, it is better to have a small $l$. It makes sense as for the performance of node classification task adjacent nodes should have similar embeddings. On contrary, for the link prediction task a high $l$ is better as a taller tree can capture inherent graph structure better which leads to a better link prediction performance. We illustrate the effect of $l$ in Figure 3.

**Effect of sibling similarity coefficient, $\gamma$:** For sibling similarity in our work, we introduce a hyperparameter $\gamma$ as positive similarity weight. We set this parameter based on the ratio between hierarchical pairs and sibling pairs to offset any respective bias. We illustrate the effect of sibling similarity coefficient, $\gamma$ in Figure 3.

We report the best hyperparameters for each dataset and task combination in Table 9 in the appendix.

## 5 Conclusion and Future Work

NODE2BINARY is the first work to propose a combinatorial optimization approach in binary space for general graph embedding. In summary, it uses a fast community detection algorithm to convert a general graph into a hierarchy partition tree and then solves an NP-hard CO problem with an innovative combination of discrete gradient descent and randomization. As our model has fewer parameters than other models (such as random-walk based node2vec) the optimization process is less costly. As a future work we plan to compute Hierarchy Tree and Sibling gradients in parallel which can speed up algorithm run time. Our experimental results demonstrate that node2binary achieves competitive results at low dimensions on standard evaluation tasks with popular real-world graphs. Thus node2binary is also well suited for applications requiring efficient energy or memory usage, such as edge devices. Experimental results also show that node2binary can be scaled to millions of nodes and be competitive with the methods designed for scalability. However, node2binary is a transductive model that cannot generate embeddings for unseen nodes in a graph. Another possible future direction is to extend our algorithm for inductive learning tasks.

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

# A  Appendix

## A.1  Reproducibility

Our code and datasets are publicly available at the following link: https://anonymous.4open.science/r/node2binary-C952/. It is implemented in Python with the PyTorch library. Although node2binary is a randomized algorithm, we maintain the same random tree $\mathcal{T}_G$ for each run to increase consistency. We also provide a command-line option --seed in our implementation to seed the random number generator and make the result deterministic.

## A.2  Computation of Gradients

The discrete gradient functions we use for updating our embeddings are designed to guide the model towards fulfilling the tree and sibling constraints. We can think about the gradient as encouraging or discouraging flipping certain bits. For positive tree pairs $(c, p)$, for each bit $j$, we want to avoid having $c_j = 0$ and $p_j = 1$. Therefore, for each pair of vectors $\mathbf{c}, \mathbf{p}$ with $c$ a child of $p$, we assign gradient values to encourage corresponding bit pairs $(c_j, p_j)$ to flip away from $(0, 1)$. If $(c_j, p_j) = (0, 1)$, then we set the gradient of both bits to 1, so that at least one of them flips. If $(c_j, p_j) = (0, 0)$, then we are fine, but we want to *avoid* flipping $p_j$, which would create a $(0, 1)$ pair. The other cases are similar and shown in Table 5.

A similar logic holds for negative samples, i.e. pairs $(c', p')$ where $c'$ is *not* a child of $p'$. In fact, the gradient for this case is exactly the negative of the gradient for positive pairs $(c, p)$, as shown in Table 6. In a similar way, we show the gradients for sibling loss, which aims to make the two bits equal, in Table 7. Note that, when $a_j \neq b_j$, we set both gradients to 1, because we don't care which bit flips, only that one of them does. This means there is a chance *both* flip, causing $a_j \neq b_j$ again. Our solution is the randomized flip function in Eq. 9, which, on average, flips at most half of the bits in the model.

**Table 5: Logic truth table for positive tree loss gradient**

| $\mathbf{c}_j$ | $\mathbf{p}_j$ | $\Delta_{\mathbf{c}_j} Loss^+$ | $\Delta_{\mathbf{p}_j} Loss^+$ | Comments |
|---|---|---|---|---|
| 0 | 0 | 0 | −1 | Don't flip $\mathbf{p}_j$ |
| 0 | 1 | 1 | 1 | Flip either (or both) bit |
| 1 | 0 | 0 | 0 | Irrelevant |
| 1 | 1 | −1 | 0 | Don't flip $\mathbf{c}_j$ |

## A.3  Fine-grained Node Classification Task

We report F1 score in Figure 4. For largest dataset (YouTube), all our competitors perform poorly. For second largest (DBLP), we perform superior except at the lowest training ratio, 0.1. For BlogCatalog

**Table 6: Logic truth table for negative tree loss gradient**

| $\mathbf{c}'_j$ | $\mathbf{p}'_j$ | $\Delta_{\mathbf{c}'_j} Loss^-$ | $\Delta_{\mathbf{p}'_j} Loss^-$ | Comments |
|---|---|---|---|---|
| 0 | 0 | 0 | 1 | Flip $\mathbf{p}'_j$ |
| 0 | 1 | −1 | −1 | Don't flip |
| 1 | 0 | 0 | 0 | Irrelevant |
| 1 | 1 | 1 | 0 | Flip $\mathbf{c}'_j$ |

**Table 7: Logic truth table for sibling loss gradient**

| $\mathbf{a}_j$ | $\mathbf{b}_j$ | $\Delta_{\mathbf{a}_j} Loss$ | $\Delta_{\mathbf{b}_j} Loss$ | Comments |
|---|---|---|---|---|
| 0 | 0 | −1 | −1 | Don't flip |
| 0 | 1 | 1 | 1 | Flip |
| 1 | 0 | 1 | 1 | Flip |
| 1 | 1 | −1 | −1 | Don't flip |

dataset we come second to NODESIG. For PPI, our model achieves best F1 score across all training ratio. Overall our model achieves better F1 score in all datasets.

## A.4 Ablation Study

**Table 8: Ablation study for all datasets (d=512), NC = Node Classification, LP = Link Prediction.**

| | dataset | | | |
|---|---|---|---|---|
| Component | YouTube | BlogCat | DBLP | PPI |
| Hier Tree (NC) | 7.77 | 6.85 | 58.3 | 5.27 |
| Sib Similarity (NC) | 6.77 | 3.44 | 39.46 | 1.34 |
| Hier Tree (LP) | 70.28 | 68.62 | 54.04 | 60.06 |
| Sib Similarity (LP) | 50.05 | 50 | 50 | 50.05 |

Our model has two major components. First is formation of hierarchy tree and second is to minimize hamming distance among the sibling pairs in the tree. We perform an ablation study where we set $\alpha$ and $\beta$ to their default values and set $\gamma = 0$ to ignore the effect of homophily for the sibling pairs. Then we set $\alpha$ and $\beta$ to 0 and $\gamma = 1$ to ignore the effect of hierachy and only consider homophily. We set rest of the parameters to the default values. We perform this

**Table 9: Best hyper parameters (Tree depth, Sibling Coefficient) for each dataset and task combination.**

| | dataset | | | |
|---|---|---|---|---|
| Task | YouTube | BlogCat | DBLP | PPI |
| Node Classification | (4,1) | (5,1) | (3,1) | (3,1) |
| Link Prediction | (5,1) | (5,1) | (4,2) | (5,3) |

experiment for both the tasks on PPI dataset (dim 512) and report the result in Table 8.

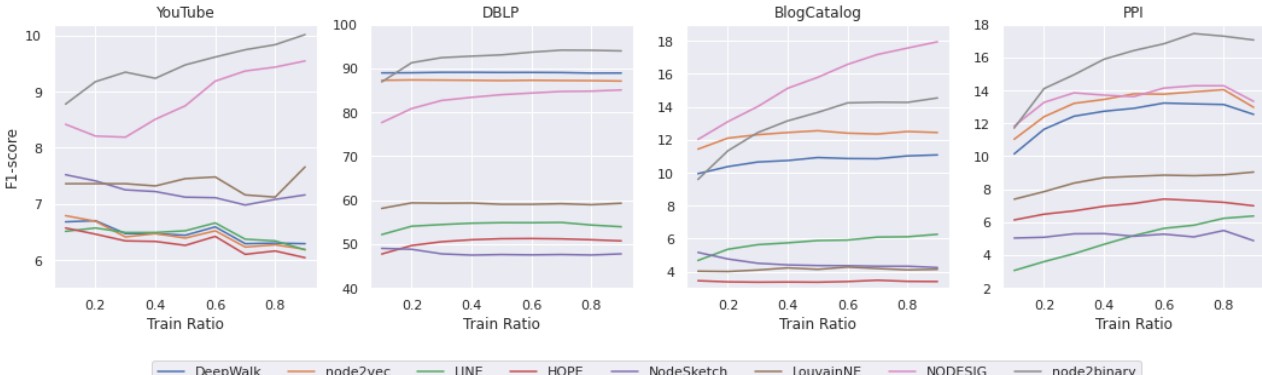

**Figure 4:** *Node Classification* experiment by varying train ratio from 0.1 to 0.9 with embedding dimension set at 512.)

