# OpenReview forum: "node2binary: Compact Graph Node Embeddings Using Binary Vectors"
_ACM.org/TheWebConf/2025/Conference — WWW 2025 Oral_

### Official Review · Reviewer_iQpv · 2024-11-11

**Novelty:** 4
**Technical Quality:** 4

**Review:**

This paper proposes an approach termed NODE2BINARY to represent nodes in networks with binary vectors, aiming to reduce storage overhead. Against the backdrop of a dramatic increase in data scale, the authors hope to replace the traditional double-precision node embeddings with low-dimensional binary vectors.

Strengths:

1.	This paper constructs a hierarchical structure based on the community, making the process intuitive and easy to understand.

2.	The authors have selected four datasets to conduct a variety of experiments, thoroughly demonstrating the performance of the node2binary approach.

**Questions:**

Weakness:

1.	The motivation for using binary embeddings to represent nodes is not clearly articulated. Binary embeddings are not an innovation of this paper; NODESIG also uses them. The main difference is that NODESIG just uses random projection while NODE2BINARY uses hashing. However, the authors do not clarify the advantages of hashing over random projection.

2.	The effectiveness of the proposed method is limited. In detail, it achieves optimal performance on node classification tasks only 50% of the time. And it is not much ahead of the best baseline.

3.	The efficiency claims are not fully substantiated. Table 4 shows the running time of node2binary and two baselines. While it is faster than NODESIG, the performance metrics (AUC for link prediction and F1-score for node classification) do not show a clear superiority over NODESIG.

4.	The experiments should align more closely with the stated motivations. The authors emphasize the memory usage reduction of binary embeddings but do not provide a comparison of memory usage between their method and the baselines.

**Reviewer Confidence:**

3: The reviewer is confident but not certain that the evaluation is correct

**Scope:**

4: The work is relevant to the Web and to the track, and is of broad interest to the community

---

### Official Review · Reviewer_ei7U · 2024-11-30

**Novelty:** 6
**Technical Quality:** 5

**Review:**

This paper focuses on graph embedding with the target of reducing the size of the embedding storage footprint. To achieve this goal, the authors do not stick to the mainstream graph neural network approach but rather convert the given graph into a hierarchical partition tree and then compute embedding of graph vertices in binary space by solving a combinatorial optimization (CO) task over the tree edges. The problem itself and the methodology proposed by the authors are novel and of practical interest. The authors have conducted extensive experiments on real-world datasets of various sizes, and the performance seems to be promising. The paper is well structured and easy to follow.

However, there are still some concerns:

1) F1 scores of all methods, including the proposed methods, are below 30 on all datasets except DBLP. Although the proposed node2binary overperforms other baselines, a low F1 score may suggest that most compared models do not work well on node classification task. This raises concerns about the evaluation. Moreover, on the DBLP dataset, the proposed node2binary performs worse than DeepWalk, at smaller bits (128 and 256), but handling this should be the strength of the proposed node2binary. A similar situation exists for the link prediction task. Almost all methods in PPI and DBLP have AUCs below 70%. However, link prediction on the YouTube and Blogcatalog datasets demonstrates the superiority of the proposed node2binary.

2) The selected non-binary baselines are not the latest. Choosing the state of the art non-binary baselines can showcase how well the latest methods can perform on these tasks. Their inability to achieve higher performance levels, would strengthen the argument in favor of the proposed method. Even if state of the art approaches are superior to the proposed method, the fact that the embedding they generate takes up more memory will not significantly reduce the paper's contribution, but it will give the reader a better idea of how much accuracy has been sacrificed to reduce memory footprint.

3) On DBLP (link prediction), the proposed node2binary performs poorly compared to other datasets. Analyzing the reason for this observation is meaningful as this may indicate why the proposed node2binary does not work well on specific types of networks. Such an outcome is perfectly acceptable (or even expected) given the diversity of the networks, but it needs to be more clearly presented and analysed.

Below are some minor problems and suggestions:

1) Running times need to be presented more clearly. 0.41x and 1.12x running times are not sufficient to allow the reader to understand the difference, as the reader does not know the base.

2) How the bits of non-binary vectors are calculated needs to be explained further.

3) Try to visualise the generated embedding, so that different methods can be analysed from another (i.e. qualitative) perspective.

**Questions:**

- The detailed explanation of how bits of non-binary vectors are calculated is missing.
- F1 scores are low for node classification on all detests except DBLP. Will this affect the comparison of methods?
- What are the reasons for poor performance of node2binary on DBLP (link prediction) compared to the other datasets?
- More details about running time of different methods should be also included.

**Reviewer Confidence:**

3: The reviewer is confident but not certain that the evaluation is correct

**Scope:**

4: The work is relevant to the Web and to the track, and is of broad interest to the community

---

### Official Review · Reviewer_u9NE · 2024-12-01

**Novelty:** 2
**Technical Quality:** 3

**Review:**

The paper focuses on embedding graph nodes into binary vectors to represent node features and reduce memory usage. However, the paper has some weaknesses as outlined below:

W1. The paper suggests using binary embedding to reduce memory usage. However, a question arises: could traditional node embedding achieve lossless compression to serve the same purpose?

W2. Why restrict the embedded vector to binary form? Would an integer vector with each dimension as an integer still be effective? The discussion on motivation lacks comprehensiveness in this regard.

W3. Comparing traditional node2vec methods, the embedding space is typically larger than that of node2binary. Therefore, why does node2binary outperform node2vec? Further discussion is needed to explain this results in the experiments.

W4. The paper introduces a combinatorial optimization problem for assigning binary vectors on a tree, but the problem lacks a clear definition of the problem. While the paper claims the problem is NP-hard, it lacks the inclusion of theoretical proof to support this claim.

**Questions:**

refer to W1, W2, and W3.

**Reviewer Confidence:**

3: The reviewer is confident but not certain that the evaluation is correct

**Scope:**

3: The work is somewhat relevant to the Web and to the track, and is of narrow interest to a sub-community

---

### Official Review · Reviewer_MMsf · 2024-12-02

**Novelty:** 5
**Technical Quality:** 4

**Review:**

The proposed method embeds graphnodesinaslowas128binarybits, which drastically reduces the memory footprint of vertex embedding vectors by several orders of magnitude. However, there are some issues that require further clarification from the authors:

1. The related work is not sufficiently researched. The authors should have included more recent studies in the related work section, analyzed the characteristics of their practices, and clarified the differences and connections between this work and theirs to further clarify the contribution of this paper.

2. On the node classification task, the proposed method is generally worse than the baseline model on both datasets, Blogcatalog and PPI. Similarly, on the link prediction task, the proposed method generally works worse than the baseline model on both datasets, PPI and DBLP.

3. The necessary ablation analysis experiments are lacking. The role of the components of the proposed method has not been demonstrated experimentally.

4. The contribution of this paper is summarized into two points in the Introduction, it seems that innovation is not enough.

**Questions:**

See Review.

**Reviewer Confidence:**

2: The reviewer is willing to defend the evaluation, but it is likely that the reviewer did not understand parts of the paper

**Scope:**

3: The work is somewhat relevant to the Web and to the track, and is of narrow interest to a sub-community

---

### Official Review · Reviewer_5XC6 · 2024-12-03

**Novelty:** 6
**Technical Quality:** 5

**Review:**

The method, Node2binary, uses discrete gradient descent and randomization to address the NP-hard problem efficiently. The authors demonstrate that Node2binary achieves competitive performance compared to state-of-the-art graph embedding methods in both node classification and link prediction tasks on four real-world datasets.

Pros.
1. The paper presents a unique approach to GRL by using binary vectors, which is a significant departure from traditional floating-point representations, offering substantial memory efficiency.
2. Node2binary is shown to be scalable and efficient, capable of handling large graphs and providing competitive results with a lower computational cost.
3. The authors effectively transform the combinatorial optimization problem into an unconstrained optimization problem, making it solvable using discrete gradient descent and randomization.
4. The availability of the code and datasets for reproducibility is a strong point, enhancing the credibility of the research.

Cons.
1. Node2binary is a transductive model, which means it cannot generate embeddings for unseen nodes. This might limit its applicability in dynamic or evolving graph environments.
2. The performance of Node2binary heavily relies on the quality of the community detection algorithm. The paper could benefit from a discussion on the robustness of the method against variations in community structure.
3. The paper could benefit from a more thorough theoretical analysis of the proposed method, including the convergence properties of the discrete gradient descent approach used.

**Questions:**

1. Since Node2binary is a transductive model, how do you envision extending it to handle dynamic graphs where new nodes and edges are continuously added?

**Reviewer Confidence:**

3: The reviewer is confident but not certain that the evaluation is correct

**Scope:**

3: The work is somewhat relevant to the Web and to the track, and is of narrow interest to a sub-community